# Spectroscopic analysis reveals that soil phosphorus availability and plant allocation strategies impact feedstock quality of nutrient-limited switchgrass

Zhao Hao [1✉], Yuan Wang[2], Na Ding[2], Malay C. Saha[2], Wolf-Rüdiger Scheible[2], Kelly Craven[2], Michael Udvardi [2], Peter S. Nico[1], Mary K. Firestone[3] & Eoin L. Brodie [1,3✉]

The perennial native switchgrass adapts better than other plant species do to marginal soils with low plant-available nutrients, including those with low phosphorus (P) content. Switchgrass roots and their associated microorganisms can alter the pools of available P throughout the whole soil profile making predictions of P availability in situ challenging. Plant P homeostasis makes monitoring of P limitation via measurements of plant P content alone difficult to interpret. To address these challenges, we developed a machine-learning model trained with high accuracy using the leaf tissue chemical profile, rather than P content. By applying this learned model in field trials across two sites with contrasting extractable soil P, we observed that actual plant available P in soil was more similar than expected, suggesting that adaptations occurred to alleviate the apparent P constraint. These adaptations come at a metabolic cost to the plant that have consequences for feedstock chemical components and quality. We observed that other biochemical signatures of P limitation, such as decreased cellulose-to-lignin ratios, were apparent, indicating re-allocation of carbon resources may have contributed to increased P acquisition. Plant P allocation strategies also differed across sites, and these differences were correlated with the subsequent year's biomass yields.

[1] Earth and Environment Sciences, Lawrence Berkeley National Laboratory, 1 Cyclotron Rd, Berkeley, CA 94720, USA. [2] Noble Research Institute, 2510 Sam Noble Parkway, Ardmore, OK 73401, USA. [3] Department of Environmental Science, Policy and Management, University of California, Berkeley, CA 94720, USA. ✉email: zhao@lbl.gov; elbrodie@lbl.gov

Biofuel crops have been developed as an alternative, carbon-neutral energy source, among which the perennial $C_4$ grass, *Panicum virgatum L.* (switchgrass), native to North America, can adapt to a wide range of environments[1], including those with marginal soils and low water input. However, in order to better manage and optimize this crop for biofuel production, it is important to understand the mechanisms that enable its adaptivity, and how nutrient-poor environments impact chemical composition, biomass yield and feedstock quality.

A long-standing barrier to this mechanistic understanding lies in the difficulty in characterizing plant chemical composition and quantifying plant-available nutrients at the rhizosphere. Phosphorus (P) is a critical nutrient[2], and poor P management poses a global risk for environmental sustainability and food security[3–5]. P limitation severely restricts photosynthesis and reduces $CO_2$ fixation[6], but upregulates pathways associated with organic acid/carboxylate exudation[7]. P limitation can also be associated with increasing biosynthesis of defense metabolites, such as increased lignification of cell walls[8], suggesting that changes in plant carbon allocation in response to P limitation may alter both the yield and the chemical composition of biofuel feedstocks, and therefore productivity[9]. In this light, it may be beneficial to monitor plant-available P concentration and plant chemical composition during the growth season, with the goal of improving biomass production[10] and optimizing the chemical composition for improving feedstock quality[11] through active land management, especially when growing in marginal soils[12].

Quantifying soil P available to plants is challenging, especially if attempting to do this dynamically during a growing season. Typical chemical extraction methods (e.g., Bray, Olsen, or Mehlich III) quantify only a fraction of the inorganic P pool and are typically measured in top soils prior to planting[13,14]. Although the P concentration data obtained with these methods have been widely used in the literature to represent total P availability, they are not an accurate measure of P available for plant growth. Perennial grasses such as switchgrass produce deep roots that explore and obtain nutrients and water from distinct locations deep into the soil, and these locations vary across the growing season and lifetime of a plant. Further, plants have developed a number of strategies[12,15–17] to access P from different types of soil, including the adaptive secretion of compounds such as organic acids, enzymes and siderophores which either mobilize soil P directly, or indirectly through their stimulation of the rhizosphere microbiome and symbiotic fungi[18–20]. Combined, dynamic growth of roots through a soil profile with distinct concentrations and chemical forms of P, an adaptive allocation of photosynthate belowground, and a microbiome with typically unknown capacity for P mobilization, makes predicting plant-available P a highly complex task.

Meanwhile, there has been renewed interest and some success in predicting plant nutrient levels using spectroscopic methods for remote sensing with the help of machine intelligence[21]. A variety of machine learning tools have been utilized to achieve satisfactory results with independent variables obtained, for example, by visible to near-infrared spectroscopy, to predict nutrient levels in shoots in agricultural crops[22–25], total nitrogen content of soils[26] and plant adaptive responses to stress[27]. Most of these tools are linear models such as partial least squares regression, principal component analysis, or support vector machines often with a nonlinear kernel, likely due to their inherent robustness and reduced chance of overfitting. Intuitively shoot nutrients should be somewhat correlated to soil nutrient availability. Thus, it is conceivable that a machine learning approach could predict nutrient availability by monitoring the biochemical signatures of plant shoots. However, to our knowledge, this aspect has not been well explored.

In this paper, we use a molecular spectroscopic method to determine and quantify the organic P ($P_o$) and inorganic P ($P_i$) in leaf tissue. This approach also provides important information on overall plant tissue biochemistry that can be used as multiplex-signatures of a plant's response to environmental conditions. P speciation (inorganic versus organic) can be quantified dynamically and feedstock quality for biofuel production can be inferred. We then use this tissue biochemical information to infer and evaluate plant-available P using a machine-learning model trained using a dataset from a controlled laboratory experiment. Building off this approach, we used the model to interpret plant spectral data from two field locations where contrasting available P was expected.

## Results and discussion

**Controlled sand-based laboratory experiment to evaluate plant biochemical responses to nutrient availability**. A series of experiments in sand cultures were performed to evaluate the dose-response of plant tissue chemistry to varying N and P. The chemical signatures in plant leaves varied substantially with P and N availability in the growth media, as shown in Fig. 1. Note that the absorbance data were normalized to the maximum to show the relative concentration changes on the same scale. We observed higher cellulose, lower lignin, lower lipids, and higher organic and inorganic phosphate concentrations in the leaves of plants grown in solution with closer to optimal (higher) P concentration, and increased lipid and amide concentrations in solution with closer to optimal N concentration. The cellulose/lignin (C/L) ratio was very sensitive to P concentration, showing a 3-fold increase from <150 μM to 500 μM P, but was not consistently sensitive to N concentration. Note that the P/N concentration likely fluctuated during plant growth and between the fluid replenishment, thus the values here refer to the average concentration through the growth. Since the C/L ratio is an important metric for biofuel production, we suggest that higher P concentrations would produce a higher C/L ratio for potentially increased biofuel yield. The higher relative amide concentration in plants grown in the lowest concentration of P (1 μM) compared to those in intermediate P concentrations (10-30 μM) likely reflected severe P-stress in these plants, as soluble nitrogenous compounds including amino-acids and amides accumulate in other species under P-deficiency[28,29], and consistent with our other observations[7].

P deficiency was also associated with higher lipid content, as indicated by the increased signal from carbonyl bonds, possibly related to the production of triacylglycerides as storage compounds under P limitation[7,9]. Note that while lignin concentrations increased gradually with decreasing P (there was no dose-response for N), cellulose concentrations showed a threshold effect with a large increase between 30 and 150 μM P, and these opposing responses manifested in a cellulose/lignin ratio that is highly sensitive to P deficiency, but not to N deficiency. The increase in lignin concentration under P deficiency was possibly related to induction of defense genes and defense metabolites and the overall shift to lower cellulose and more lignin may represent a more pathogen-resistant, rigid cell wall[30–33].

**Field experiment to evaluate the relationship between plant-available P concentration and switchgrass biochemistry.** Because of the strong dependence of feedstock chemical composition on soil phosphorus concentration in the controlled-growth experiments, we evaluated switchgrass growth at two field locations contrasting in soil P availability. The soil texture of these locations differs with the RR site being a sandy loam and 3rd St

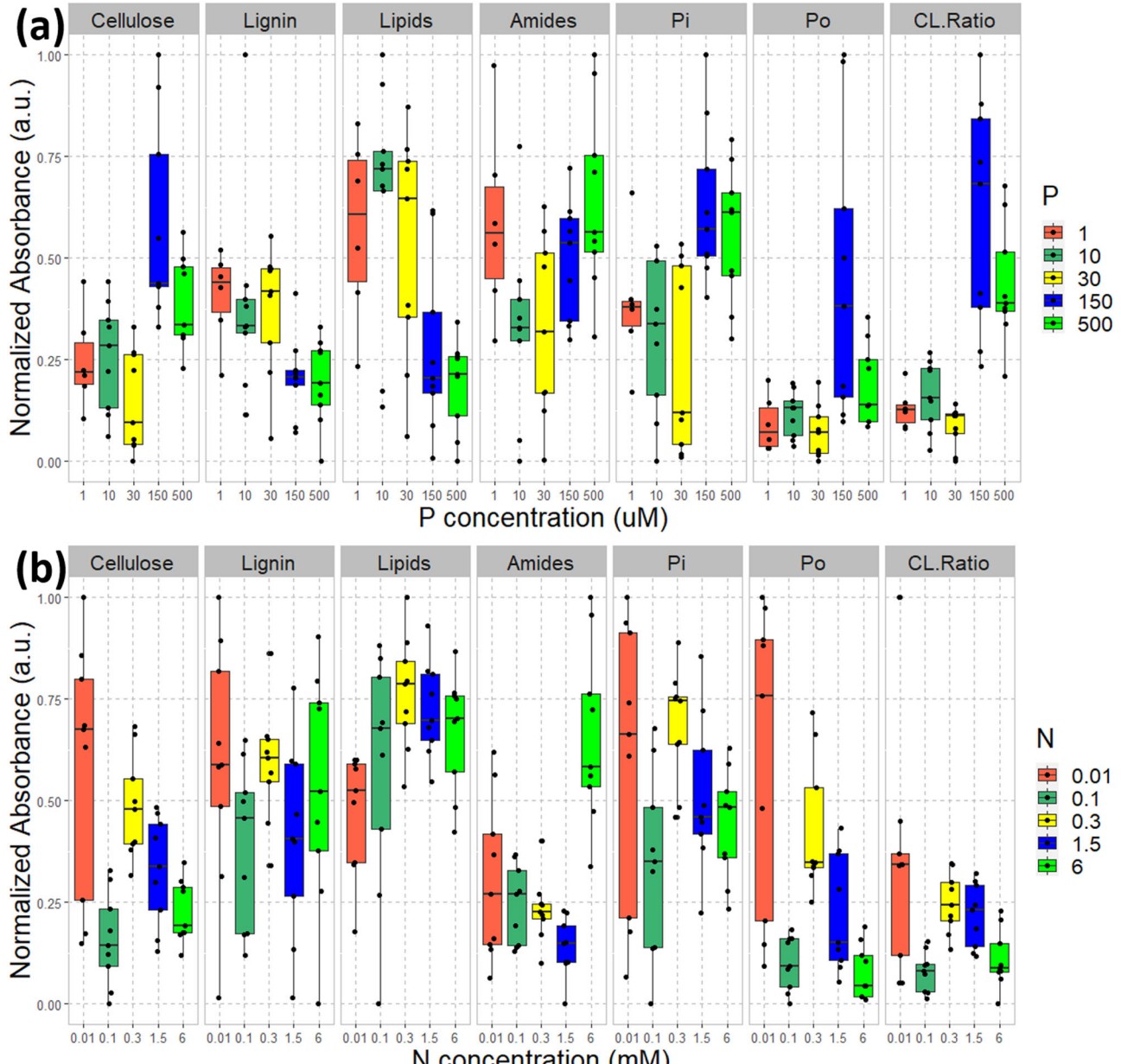

**Fig. 1 Biochemical components of leaf tissue.** Relationships between the normalized concentration (presented here as normalized absorbance at corresponding infrared frequencies) of biochemical components (cellulose, lignin, lipids, amides and inorganic/organic phosphates) in the leaf tissue samples of switchgrass plants and the concentration of **a** P- and **b** N- in the growth media of laboratory hydroponic experiments. "CL.Ratio" stands for the cellulose:lignin ratio.

being a silt loam. Chemical characterization of bulk soil samples indicated a significantly ($p = 2.2 \times 10^{-16}$) higher Mehlich-III extractable P concentration in RR soils (80 ppm) relative to 3rd St (~25 ppm) (Fig. 2a), with no significant seasonality observed ($p = 0.892$). In general, plants grew taller in the RR plot than at 3rd St, reaching maximum heights at T4.

Our leaf-tissue measurements showed that leaf $P_i$ concentration increased over the course of the growing season in both field experiments. The $P_i$ concentration of leaf tissue collected at RR was higher than that at 3rd St (Fig. 3a), consistent with the bulk soil characterization, although the plant $P_i$ as determined by FTIR was more similar than the extractable soil P data might have suggested. This may be expected due to P homeostasis and overall biomass difference. $P_i$ concentration showed an earlier increase in plants at Red River (T2), presumably reflecting greater uptake

early on due to higher concentrations of plant-available phosphate in the soil. Further increase in the concentration of $P_i$, especially in the late season along with the decrease of $P_o$ at RR may reflect mobilization of Pi for translocation elsewhere in the plant, including storage tissues that support regrow in the next growing season[34].

The trend of organic phosphates in the lower panel of Fig. 3 shows that, contrary to that of $P_i$, the concentration of $P_o$ plateaued in the later stages of growth at around T4, when the plant reached maximal biomass as indicated by the maximal plant heights (Fig. 2b). $P_o$ concentration decreased significantly during senescence (T5) in plants grown on the higher P soils at Red River. This explains the transient sharp increase in $P_i$ described above and the large decrease of $P_o$ and total P (Fig. S1). Seasonal increase of total P content in the shoots of switchgrass has been

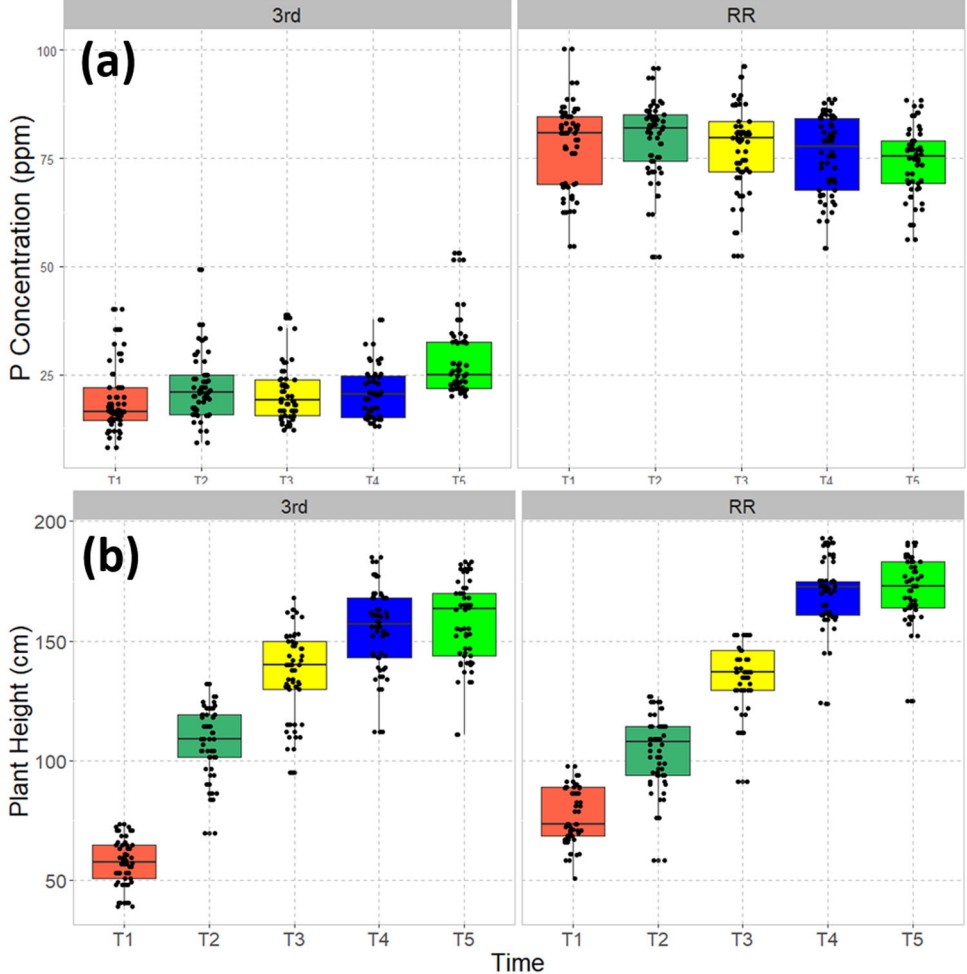

**Fig. 2 Water-soluble P concentration and seasonal change in plant height. a** The water-soluble P concentration in the soils at the two field experiments at 3rd Street (silt loam soil) and Red River (sandy loam soil), respectively, with the boxes color-coded by the harvest times here and in subsequent figures. There was no significant difference in P concentration over the growing season at either location ($p = 0.892$). Soil P concentrations are significantly higher at Red River ($p = 2.2 \times 10^{-16}$). **b** The seasonal change of plant heights at the two field plots.

observed before[35], but our spectroscopic method enabled us to dissect P speciation during the growth season. Meanwhile, we observed a similar trend in concentration of lipid signature in the late growth stage (Fig. S2). Since the leaves we collected tended to be younger leaves to be consistent with our sand-based experiments, the maximum P concentration at T4 may reflect a combination of P uptake over the growth period, plus reallocation from old to younger leaves, resulting in higher concentrations of major P-containing molecular classes like phospholipids and/or ribosomal RNA.

**Machine learning model prediction of plant-available P.** Because of the critical role of P in the growth of switchgrass and its strong correlation with biochemical composition for this biofuel species, we believe the seasonal characterization of plant-available P in the rhizosphere and P speciation may be beneficial for crop management and improved environmental outcomes. We demonstrate here that a machine-learning (ML) model can be used to quantify P availability using the plant leaves themselves as sensors.

Since the nutrient concentration in the rhizosphere in a hydroponic substrate is relatively well controlled, this experiment allowed us to develop training data for an ML model. We achieved a principal component regression (PCoR) model with a high $R^2$ of ~1 (with 41 principal components, see the learning

curves in Fig. S3), which allows us to predict plant-available P concentrations based on the spectral data collected on the leaf tissue from field-grown plants. The predicted P concentrations are shown in Fig. 4a. Note that in a more traditional approach, the model prediction would be further validated by another independent method to evaluate the model's accuracy. However, such a method for accurate estimation of bioavailable P concentration through the soil profile over time does not yet exist in practice. We believe that our model contains an accurate statistical description of the correlation between the P concentrations in the growth media and all the spectral features in younger leaf samples, given the high accuracy achieved with large concentration range and the high affinity of P uptake; thus this model can be used for prediction of the P concentration available to each plant within the rhizosphere.

The predicted P concentration available to the plants shows a gradual increase and then a sharp dip in T4 when the plant reached maximal biomass, reflecting an increase in P uptake at T4 and a quick decrease in P uptake at T5, when the shoot senesces. As a perennial plant, switchgrass remobilizes and stores P in roots to support the subsequent year's growth, consistent with previous observations[34] with P remobilization efficiency ranging from 31% to 65% in different ecotypes. The increase in $P_i$ in the tissue at the later stage of growth, the strong correlation of cellulose content with total P concentration, and the large reduction of plant-

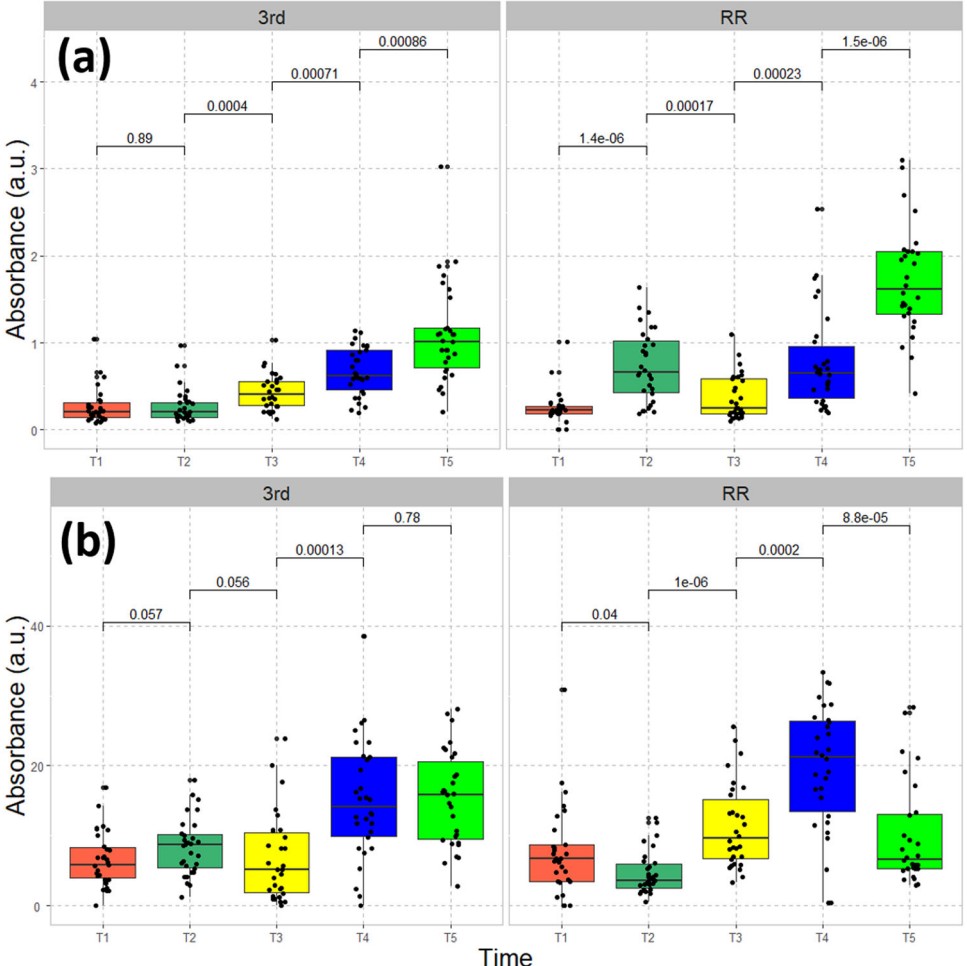

**Fig. 3 Seasonal dynamics of P from leaf-tissue samples. a** Inorganic and **b** organic P in leaf-tissue samples taken over the growing season from the two field experiments at 3rd Street and Red River. *P*-values derived from pairwise comparisons are shown on the horizontal lines.

available P in the rhizosphere provided us with a clear picture of the interaction of P availability and tissue composition during the life cycle of switchgrass. This may be a consideration in the timing of harvest to achieve optimal biofuel yield and reserve P in the root for the optimal growth in the next year.

The total P concentration in the roots followed a similar seasonal trend, showing a reduction of total P concentration near the period of maximal growth and at least a partial recovery in T5. The P concentrations measured in roots by ICP-MS at T5, a point at which plants had begun senescing, were similar across the two field locations ($p = 0.6626$), with a mean value of 729 ppm at 3rd St and 703 ppm at RR, respectively. The similar root P concentrations at T5 may be reflective of reduced plant P demand during senescence as well as plant P re-allocation (Fig. S4) and is described further below. Note also that the root samples we collected are a small fraction of the whole root system, and switchgrass is known to develop large root crowns for nutrient storage.

As mentioned, the inferred plant-available P concentrations (and the measured root P) across these two field locations, with distinctly different extractable soil P, were surprisingly similar throughout the growing season. This disconnect, suggests that extractable P from soils, though sampled near the roots, do not reflect true P availability to roots, or that plant adaptation to P limitation (direct or through associated microorganisms) changes the fraction of P that is in fact plant-available as the plant develops. This latter

possibility could be explored by assessing the ratio between the predicted plant-available P concentrations and the extractable soil P concentrations, as shown in Fig. S5b. This shows that plants at 3rd St. "experienced" a comparatively higher concentration of P than would be expected based on soil chemical extractions and the plant growth rates correlated better with the plant-based estimates of P availability, pointing to important biological, e.g., plant and microbial, processes that liberate P associated with soil minerals or organic matter in this higher clay content soil.

Given the greater availability of P at RR, one might expect less need for plants to deploy adaptive strategies to obtain P, and possibly a tighter connection between seasonal availability of P and plant height at RR compared to those at 3rd St, which was indeed the case (Fig. S5a). Reduced energy costs associated with P-acquisition, in addition to higher P availability and other factors, may explain the greater biomass of plants grown at RR compared to those at 3rd St (with average mass of 333 g and 208 g per plant, respectively; $p = 2.2 \times 10^{-16}$).

The increase in P concentration in the late season in roots and the speciation of $P_o$ and $P_i$ in leaves indicate that switchgrass plants increase P mobilization and storage during shoot senescence, as discussed above. We further explored this point and observed a clear association between second-year biomass production and the ratio of $P_i/P_o$ in the shoots at senescence (Fig. S6). This suggested that effective reallocation of P (i.e., a higher $P_i/P_o$ ratio in the late season) during senescence may

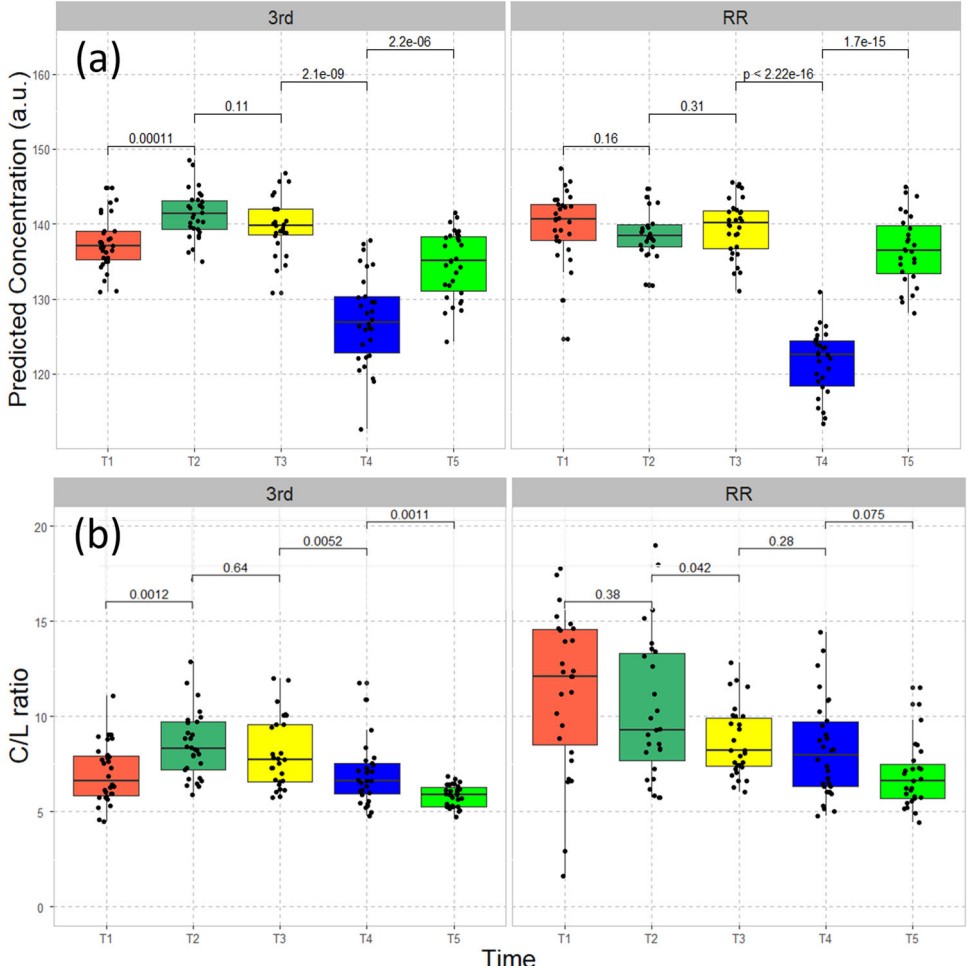

**Fig. 4 Machine learning predictions of seasonal P concentration and cellulose/lignin ratio of leaf-tissue samples. a** The seasonal dynamics of P concentration available to plants based on a machine learning analysis of leaf tissue spectral properties from the two field experiments at 3rd Street and Red River, with the p-values from two-sample tests labeled above the boxes. *P*-values derived from pairwise comparisons are shown on the horizontal lines. **b** Cellulose/lignin (C/L) ratio of leaf-tissue samples collected from the two field experiments at 3rd Street and Red River. Overall, the ratios are significantly higher (*p* = 0.0023) in Red River than those in 3rd Street. *P*-values derived from pairwise comparisons are shown on the horizontal lines.

contribute to the increased total biomass observed in second-year growth ($p = 4.4 \times 10^{-16}$).

Switchgrass plants grown at the RR had a higher cellulose/lignin ratio than those of 3rd St ($p = 0.0023$), coincident with higher extractable soil P concentrations at RR. Furthermore the cellulose/lignin ratio decreased over time as plants matured in both plots (Fig. 4b). These observations are consistent with the observations in the hydroponic experiment, and with the development of harder tissues later during development of perennial plants[2]. Taken together, these results show that limitation of P availability and the resulting imbalance between supply and demand of P for growth contribute to reduction of total biomass and the feedstock quality.

**Conclusions.** In conclusion, we designed two sets of experiments, one in a controlled laboratory hydroponic setting and the other as a field experiment across locations with contrasting soil properties that influence nutrient availability. This allowed the estimation of integrated plant-available P through the root zone based on a ML model trained using the laboratory data, with a prediction further supported by chemical analysis of roots. We observed a series of consistent biochemical changes in shoot tissue biochemistry when the plants were grown in a low-P environment, including a large decrease in cellulose/lignin ratio,

decreased lipids, and correlated changes in amide concentrations. We observed a similar biochemical shift in shoot tissue from plants grown in the field sites with lower extractable soil P, which leads to our conclusion that P availability during plant growth strongly impacts cellulose/lignin ratio, an important metric for feedstock quality.

The ML methods we developed allowed us to observe the seasonal dynamics of P availability in the rhizosphere. In parallel we show evidence for differential P reallocation within leaf tissue, as well as differential recovery of total P in root tissue late in the growing season. A positive correlation between the successful translocation of P and the total dry-mass production in the second year, highlights the critical role of P in the sustainability of feedstock growth as well as chemical quality. Furthermore, despite the two field sites showing significant differences (almost 3x) in extractable soil P near the roots, plant height was surprisingly similar; further our plant-based sensing of plant-available P concentration suggested that plants at 3rd St accessed pools of P not accurately represented by typical soil extracts. By accessing alternate pools of P (not represented by chemical extractions) the plant incurs several costs including decreased carbohydrate production[7], and as we have shown, other changes in tissue biochemistry. This adaptive capacity for nutrient acquisition has consequences for not just feedstock yield but also

feedstock quality that will influence the yield of cellulosic-derived bioproducts. The mechanisms underlying this adaptive capacity are not well understood and could be a target for enhancement. Overall, we believe that the characterization method developed here is amenable to high throughput assessment of bioenergy feedstock biochemistry and may prove useful in guiding customized nutrient amendment regimes to improve feedstock yield and quality.

## Methods

**Switchgrass plants**. Switchgrass cultivar Alamo seeds used in this study were produced at the Noble Research Institute (NRI, Ardmore, Oklahoma, USA). They were surface-sterilized by a 2-min treatment with 70% ethanol, de-husked in 60% $H_2SO_4$ for 30 min, followed by a 30-min treatment with 50% Clorox® (8.25% sodium hypochlorite, Clorox, Oakland, CA, United States) containing 0.1% TWEEN 20 (AMRESCO, Solon, OH, United States) and five rinses between each step with sterile water. They were germinated at 28 °C for 5 d on sterile, wet filter paper in a dark environment.

**Laboratory-based nutrient limitation experiments and sample preparation**. A series of sand-based hydroponic experiments were performed to establish the ground-truthing correlation between the tissue chemistry and the available P, N concentration. The details of these experiments have been reported by us previously[7]. In short, we transplanted seedlings of comparable size into growth cones which were filled with acid-washed all-purpose sand mix. A total of 78 plants were watered daily to field capacity during a 4-week growth period, half of which with a nutrient solution containing 1, 10, 30, 150, or 500 μM of $P_i$ (with optimal 6 mM of N), supplied as $KH_2PO_4$, and the other half with 0.01, 0.1, 0.3, 1.5, or 6 mM of N (with 500 μM of P), supplied as $KNO_3$. There were 9 replicates for each condition, except that three of the plants grown with $P_i = 1$ μM had died before the harvest, which were not included in the analyses. The plant samples for chemical analyses were rinsed in Milli-Q water, blotted dry, immediately frozen by liquid nitrogen and stored at −80 °C before freeze-drying and grinding. The samples for biomass measurement were dried at 65 °C in pre-weighed paper bags until a constant weight was achieved.

**Field experiments and sample preparation**. Field experiments were conducted at two locations managed by the Noble Research Institute in Ardmore, Oklahoma. These locations are referred to as "Third Street" (3rd St, latitude: 34.172100 N and longitude: −97.07953 W) with a silt loam textured soil and "Red River" (RR) near the Oklahoma-Texas border (latitude: 33.8820278 N and longitude: −97.2755056 W) with a sandy loam textured soil (Figure S7). In May 2016, we planted Alamo seedlings across both fields. Previous studies have shown the Alamo population has a large biomass yield variation resulting from different genetic backgrounds. We randomly selected 30 plants per plot for continuous sampling and growth data collection over 1 year, roughly every month for a total of five time points, corresponding to early vegetative growth in June (T1), late vegetative growth in July (T2), reproductive growth between August and September (T3), maximal biomass in October (T4), and the senescence period in November (T5), respectively.

For each switchgrass plant, soil cores (15.24 cm or 6 inch deep and 6.35 cm or 2.5 inch in diameter) were taken adjacent to the plants. Roots collected from the soil cores were rinsed with PBS buffer, freeze-dried, and powdered for elemental analysis (P and N) with inductively coupled plasma-mass spectrometry (ICP-MS). Soil samples taken near the roots of the plants were analyzed by the Mehlich III method[36] to quantify the concentration of phosphate and nitrate. This extracting solution consists of multiple chemical solutions, including acetic acid, ammonium nitrate, ammonium fluoride, nitric acid, and the chelator, EDTA. ICP-MS was then used to determine water-soluble P in the soil.

All leaf samples were freeze-dried and ground for the aforementioned chemical analyses. Samples for total-dry-mass measurement were dried at 65 °C until a constant weight was achieved.

**Attenuated total reflection—Fourier transform infrared (ATR-FTIR) spectroscopy**. The ground leaf samples were measured directly by an ATR-FTIR spectrometer (Nexus iS50 spectrometer with Smart iTR ATR accessory, Thermo Fisher Scientific)[37], with 32 averaging scans and a spectral range from 4000 to 600 cm$^{-1}$ with a resolution of 4 cm$^{-1}$. Each sample was pressed down to contact the surface of a Ge crystal. A portion of evanescent infrared waves was absorbed at the interface, and the internally reflected photons were then collected by a deuterated triglycine sulfate (DTGS) detector to acquire an FTIR spectrum. The penetration depth at the Ge/sample interface is on the order of tens of micrometers, which makes it possible to obtain leaf chemistry in a confined nanoliter volume.

**Infrared signatures used in this study**. The phosphate group absorbs light strongly at ~1000 cm$^{-1}$, which includes three degenerate symmetric and asymmetric vibrations. When it forms a bond with the other species, such as with inorganic polyphosphate and organic phosphorus compounds[38,39], the peaks are separated into frequencies covering from ~1400 cm$^{-1}$ to ~800 cm$^{-1}$, two of which are of particular interest in this context, because of their distinctive locations from the C–O–C vibrations, mostly from polysaccharides, which are marked by a broadband absorption around 1000 cm$^{-1}$. Here, we focus on a sharp peak related to phosphoryl group (P=O stretch) at ~1200 cm$^{-1}$ and another sharp feature, albeit being weaker in strength, related to P–O–H and P–O–C deformation at ~980 cm$^{-1}$. Our quantum chemistry simulation of two model phosphorus compounds (a phosphoryl chloride molecule for $P_i$ and a glucose-6-phosphate molecule for $P_o$, Fig. S8) confirmed our assignments as referenced in the literature[40–44]. Additionally, we assigned the signature peaks at ~1510 cm$^{-1}$ (aromatic C=C) for lignin, ~1550 cm$^{-1}$ (Amide II) for amides, ~1160 cm$^{-1}$ (C–O–C, ether linkage) for cellulose and ~1710 cm$^{-1}$ (carbonyl) for lipids[45–47], respectively. The individual peaks were analyzed with a model consisting of a number of oscillators with the least squares regression. The absorbance, or the derived peak area, follows Beer's law as proportional to the molar concentration of the corresponding chemicals[41]. In this context, we used $P_i$ to represent the inorganic polyphosphates derived from the phosphoryl group, and $P_o$ to represent the organic phosphates (or organophosphorus) from the P–O–H/P–O–C deformation (Fig. S9).

**Ion-exchange chromatography (IC)**. We cut a fresh leaf into two small pieces across the vein and dried them in an oven at 65 °C, resulting in about 30 mg in dry weight. The sample was homogenized into fine powder with 1 mm glass beads at 30 revolutions/second for 2 min in TissueLyser II (QIAGEN). We then weighed out 5 to 6 mg of the powder sample, and mixed it well in 1.5 mL Milli-Q water. The mixture was then incubated for 1 h, sonicated for 20 min, filtered with a 0.2 μm filter tip, and submitted for IC analysis (Dionex ICS-5000 plus, Thermo Fisher Scientific) to quantify total phosphates. An AG11HC guard column was used along with chromatographic separation using a Dionex CS12A, Ion Pac ($2 \times 250$ mm) analytical column at 20.5 °C, and an injection volume of 25 μL. The elution of anions was achieved with a concentration gradient of 6 mM to 21.5 mM in 16.5 min, 21.5 to 60 mM in 6.5 min, and at 60 mM for 3 min, then re-equilibrated at 6 mM for 8 min at a flow rate of 0.33 ml/min. Standard anions (Dionex, Thermo Fisher Scientific) were used, with ion quantification using commercial software (Chromeleon 7.2 SR4, Thermo Fisher Scientific).

**Statistical model and reproducibility**. We built a PCoR model[48] with 41 principal components (PCs) for P and 44 PCs for N, respectively, based on training spectra with bootstrapping (a total of 6000 random samples with replacement) obtained from the 78 plants in the laboratory-based experiment, to predict the N, P concentration values in the growth media from the baseline-corrected spectral data. Each plant sample was split for three separate FTIR measurements for quality control purposes; the standard deviations of these spectra at each nutrient condition were shown in Fig. S9a. We believe that the number of independent plants in the laboratory-based experiment is adequate for proper training of this linear model, although a larger number of independent plants with additional nutrient-limit conditions would improve the prediction accuracy in the field and its reproducibility in large scale applications because of the law of large numbers. The optimal number of components for the PCoR model was selected through extracted PCs to obtain the lowest cross-validation error from a 10-fold cross-validation (mean squared error of prediction, MSEP < 0.01). A k-fold cross-validation starts with a random partition of data in k (k = 10 in our case) parts or folds. We train the model on the k-1 folds and then validate with the one-fold. This process is then repeated k times until each fold has been used for testing once to prevent overfitting[49]. A similarly high accuracy was achieved on the N prediction as well, although the N availability is not our main focus in this research given it's not solely dependent on the uptake from soil. We used statistical functions included in base package of R for the two-sample student t-tests and the multivariable analysis of variance tests[50], and add-on packages for spectral data processing (hyperSpec)[51], PCoR (pls)[52] and plotting (ggplot2 and ggpubr)[53].

**Reporting summary**. Further information on research design is available in the Nature Research Reporting Summary linked to this article.

## Data availability

All data generated and analyzed in this study are included in the paper, Supplementary information, and Source data file (Supplementary Data 1).

## Code availability

The code used for the PCoR model training and validation can be found at ref. [54].

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

## Acknowledgements

This research was supported by the U.S. Department of Energy Office of Science, Office of Biological and Environmental Research Genomic Science program under the award number DE-SC0014079 to the UC Berkeley, Noble Research Institute, University of Oklahoma, the Lawrence Livermore National Laboratory, and the Lawrence Berkeley National Laboratory. Part of this work was performed at Lawrence Berkeley National Laboratory under contract DE-AC02-05CH11231.

## Author contributions

Z.H. and E.L.B conceived and developed the initial concept. E.L.B., M.U., and M.K.F. proposed and established the collaboration and guided the overall project direction. All authors contributed in data analysis and interpretation in experiments they conducted. Y.W., M.C.S., and K.C. coordinated field experiments, sample preparation and data

collection. N.D. and W.S. coordinated the laboratory-based experiments, sample preparation and data collection. Z.H. reviewed and statistically analyzed all data. Z.H. and E.L.B. wrote the first draft of the manuscript in coordination with Y.W., N.D., P.S.N., M.C.S., W.S., K.C., M.U., and Z.H. revised and finalized the manuscript with inputs from all authors. All authors approved the final manuscript.

## Competing interests

The authors declare no competing interests.
