## [Peer Review File · Communications Biology]

Reviewers' comments:

Reviewer #1 (Remarks to the Author):

(Please also see attachment for visualization provided by reviewer)

The manuscript is very well written. They have tried to identify the biochemical changes in leaves under P deficient conditions in Switchgrass (*Panicum virgatum* L.). This is a very fundamental study supported by good dataset. But I do have some suggestions which are provided below
Keywords are missing

Line no. 106-108: Add north and west for latitude and longitude

Line no. 118: Full form of "PBS" at its first appearance

What was the range of the FTIR spectrometer (e.g. 4000-400 cm⁻¹)? Please mention that. The spectra were captured 4 cm⁻¹. Whether it was resampled to get the spectra at 1 cm⁻¹ or the raw spectra was directly used for PCR model building. Mention these in details.

What was the dependent and independent variable for PCR model? Please mention that clearly.

44 PCs seem very high which may lead to model overfitting. If you are using "pls" R package, you can use "selectNcomp" function for choosing the best number of components.

Whether the PCoR model was validated with independent samples. What was the real sample size?

Line no. 195: absorbance or concentration?

How were the cellulose, lignin, lipids, amides measured in the leaf samples? These should be mentioned in the materials and methods section.

Line no. 269: In the materials and methods section, it has been mentioned as 44 PCs. Please check it.

Line no. 269: The R² is from calibration dataset with very high number of PCs indicate overfitting. Divide the total dataset in 2 parts (2/3rd and 1/3rd). Use 2/3rd dataset for model building or calibration i.e. obtaining optimum number of PCs and then validate the PCoR model with remaining 1/3rd data. You will see that with independent data, the R² will decrease significantly.

Provide proper table reference (e.g. total P concentration in root, predicted P concentration available to plants) for every paragraph.

Figure S2: Show the mean infrared spectra with SD for all the samples analysed in this study (Reference: Fig. 1. of Das, B., Manohara, K.K., Mahajan, G.R. and Sahoo, R.N., 2020.

Spectroscopy based novel spectral indices, PCA-and PLSR-coupled machine learning models for salinity stress phenotyping of rice. *Spectrochimica Acta Part A: Molecular and Biomolecular Spectroscopy*, 229, p.117983.

Figure S5: From the figure of R² vs. number of components, it is very clear that after PC number 15 there is very small change in the R². So, the optimum number of PC should be 15.

Figure S6: Whether the data for T1 is not available for RR site. Please mention it in the figure title.

How far it is acceptable to compare one treatment to other treatment of a same site? I think it should be better to compare 2 locations for the same treatment (e.g. T1).

Like following

Reviewer #2 (Remarks to the Author):

The paper claim the use of new analytical procedures including learning-assisted spectroscopic and machine learning to predict plant available phosphorus based on lab data in order to interpret plant mineral spectra data under field conditions of P nutrient limitation.

The manuscript is novel and with major interest for the soil & plant biology community.

Main weakness of this paper:

I guess you should avoid the ion confounding factor, typical of plant mineral nutrition experiments. Please have a look to the next paper: Niedz, R. P., and Evens, T. J. (2006). A solution to the problem of ion confounding in experimental biology. *Nat. Methods* 3, 417–417.doi: 10.1038/nmeth0606-417).

The problem is you have supplied P and N as salts with also contain K (see lines 99-100). Then authors should taking into account the total amount of K added (supplied in both salts).

I also strongly recommend to analyze the K content together with P and N (Line 119). If possible, re-analyze and add into the ML model, to predict the N, P and concentration and effects (Lines 179-180 and 191-200).

About the ML Models (lines 272-275) I am not expert in that one the authors used, but I guess there other based on different AI algorithms such as artificial neural networks, which include several statistical fitness criteria to obtain the best train Set R to validate the predictions. For example, Cross-Validation which split the data into subgroups that can be used for training, other for test and finally, others unseen for the model, to validate. Have you split your data in those groups or used all for training and/or testing? May be you can used other (there are several) AI algorithms for model and in silico validate your model.

Also, did you carried out any statistical method (such as ANOVA) to verify that there are not statistical differences between the experimental and predicted values in order to assess the model accuracy? (see lines 276-279). If done, please include those data in the manuscript. In this sense, I miss a table with the training parameters used for the model (currently not included).

I suggest to explore the use of other ML models, based in Artificial Neural Networks, to model these results and establish what ion was/were key (P or N or K) in the parameters detemined (cellulose, lignin, lipids and so on, content).

The rest of the papers is excellent and worthy to be published.

Minor issues:

Line 117. Please use SI units instead "inch" (such as centimeters or milimeters)

Reviewer #3 (Remarks to the Author):

The manuscript presents a study on the prediction of P content using spectroscopy and machine learning (Principal Component Regression). This is a novel and solid scientific work and I have only two minor suggestions. First, it would be useful to include in the introduction some background regarding the use of Machine Learning techniques for nutritional analysis. Investigations on the subject are aplenty and have yelded some very interesting results. Since this is not a review article, I would not expect an in-depth analysis, but some general comments on the state of the art would be useful. Second, it would be interesting if the authors could speculate a little about the generality of the results obtained. Can we expect that the results would be the same if the study was carried out in areas with different climate and soil characteristics?

Reviewer #1 (Remarks to the Author):

(Please also see attachment for visualization provided by reviewer)

The manuscript is very well written. They have tried to identify the biochemical changes in leaves under P deficient conditions in Switchgrass (*Panicum virgatum* L.). This is a very fundamental study supported by good dataset. But I do have some suggestions which are provided below

[Author's Response] We appreciate the great comments from all the reviewers! We take our study and the reviewers' comments seriously and we are grateful for these constructive suggestions which have greatly improved our work.

1.

Keywords are missing

Line no. 106-108: Add north and west for latitude and longitude

[Author's Response] We've added the above information, and we will select keywords when the manuscript is accepted.

2.

Line no. 118: Full form of "PBS" at its first appearance

[Author's Response] Full form of PBS has been provided.

What was the range of the FTIR spectrometer (e.g. 4000-400 cm⁻¹)? Please mention that. The spectra were captured 4 cm⁻¹. Whether it was resampled to get the spectra at 1 cm⁻¹ or the raw spectra was directly used for PCR model building. Mention these in details.

[Author's Response] We've updated to include the spectrometer range and resolution. Raw spectra after baseline correction were directly used for the PCoR model building.

3.

What was the dependent and independent variable for PCR model? Please mention that clearly.

[Author's Response] We use spectral data as inputs, so infrared frequencies are used as independent variables, with nutrient inputs as dependent variables. We have now mentioned that in this revised version.

4.

44 PCs seem very high which may lead to model overfitting. If you are using "pls" R package, you can use "selectNcomp" function for choosing the best number of components.

Whether the PCoR model was validated with independent samples. What was the real sample size?

[Author's Response] The model was validated through a 10-fold cross validation with a total of 78 independent samples with 3 replicates for each measurement and a total of 313 frequency bands, which generated a total of 313 principal components (PCs). Therefore the numbers of PCs we used are relatively small. We updated the training curves in Figure S5 to include all the PCs for your further examination. We used the same principle as in the *selectNcomp* function by choosing the optimal number at the first global minimum of RMSE or maximum of R2 in the training curve.

5.

Line no. 195: absorbance or concentration?

How were the cellulose, lignin, lipids, amides measured in the leaf samples? These should be mentioned in the materials and methods section.

[Author's Response] We use absorption spectroscopy which provides absorbance representing the molar concentration following Beer's law. We discussed how to obtain these values in the methods section (lines 159 - 183).

6.

Line no. 269: In the materials and methods section, it has been mentioned as 44 PCs. Please check it.

[Author's Response] Our apologies, thank you for spotting that – there was a typo in the method section. 41 PCs were indeed used for P modeling (44 for N). We've made the change in this submission (lines 201-202, and the caption of figure S5).

Line no. 269: The R2 is from calibration dataset with very high number of PCs indicate overfitting. Divide the total dataset in 2 parts (2/3rd and 1/3rd). Use 2/3rd dataset for model building or calibration i.e. obtaining optimum number of PCs and then validate the PCoR model with remaining 1/3rd data. You will see that with independent data, the R2 will decrease significantly.

Provide proper table reference (e.g. total P concentration in root, predicted P concentration available to plants) for every paragraph.

[Author's Response] Thanks again - as stated in our response above we used a 10-fold cross validation to train the linear PCoR model with 10% test data for each iteration, which is the default of the *pcr* function. We verified the learning curve with a manual 3-fold cross validation as the reviewer suggested. The R2 value seems to be similar to what we got with the 10-fold C.V.

7.

Figure S2: Show the mean infrared spectra with SD for all the samples analysed in this study (Reference: Fig. 1. of Das, B., Manohara, K.K., Mahajan, G.R. and Sahoo, R.N., 2020.

Spectroscopy based novel spectral indices, PCA-and PLSR-coupled machine learning models

for salinity stress phenotyping of rice. Spectrochimica Acta Part A: Molecular and Biomolecular Spectroscopy, 229, p.117983.

[Author's Response] We have added these plots as suggested in the supplementary information as Figure S9.

8.

Figure S5: From the figure of R2 vs. number of components, it is very clear that after PC number 15 there is very small change in the R2. So, the optimum number of PC should be 15.

[Author's Response] Given a total of 313 principal components (PCs) available in this PCoR model, the number of PCs we used is relatively small. We chose 41 PCs in P modeling at the first global minimum of RMSE. Given the flattened curve after the chosen number of PCs in the 10-fold cross validation, we believe that we didn't overfit the model to the available dataset with rich spectral information.

9.

Figure S6: Whether the data for T1 is not available for RR site. Please mention it in the figure title.

[Author's Response] Because the roots were very young in the beginning of growth season, we didn't aggressively sample them. As a result no root data at T1 was available for the RR site. We added an explanation in the caption.

10.

How far it is acceptable to compare one treatment to other treatment of a same site? I think it should be better to compare 2 locations for the same treatment (e.g. T1). Like following

[Author's Response] This is a good suggestion. However, we are interested in the P availability and its impact on the growth pattern. It seems that although P-availability in the two field plots are NOT significantly different, the growth patterns of switchgrass over time are different. There may be a better way to describe these differences, but we chose to emphasize the p-values of comparisons between the time points rather than across sites. We hope this emphasis makes sense.

Reviewer #2 (Remarks to the Author):

The paper claim the use of new analytical procedures including learning-assisted spectroscopic and machine learning to predict plant available phosphorus based on lab data in order to interpret plant mineral spectra data under field conditions of P nutrient limitation.

The manuscript is novel and with major interest for the soil & plant biology community.

[Author's Response] The authors thank the reviewer for the positive and constructive suggestions which improved the manuscript.

Main weakness of this paper:

1.

I guess you should avoid the ion confounding factor, typical of plant mineral nutrition experiments. Please have a look to the next paper: Niedz, R. P., and Evens, T. J. (2006). A solution to the problem of ion confounding in experimental biology. *Nat. Methods* 3, 417–417. doi: 10.1038/nmeth0606-417).

The problem is you have supplied P and N as salts with also contain K (see lines 99-100). Then authors should taking into account the total amount of K added (supplied in both salts).

I also strongly recommend to analyze the K content together with P and N (Line 119). If possible, re-analyze and add into the ML model, to predict the N, P and concentration and effects (Lines 179-180 and 191-200).

[Author's Response] This is an important point, thank you for raising it and providing a reference. As KNO_3 and KH_2PO_4 were used, if the response was due to K then we would have expected similar responses but N and P responses were very different and the P response corresponded to known P containing biomolecules. So we believe our conclusion is the most parsimonious and basically we did have a control K without P (KNO_3) and K without N (KH_2PO_4). The current dataset doesn't include a large variance of K concentrations, so it would be difficult to predict the K content at the moment. However we will include this consideration in future work in a different field setting as this current project has ended.

We did include a note, in the revised version of supplementary figure S5, about the possible ion confounding effect between K and NO_3^- , based on the reference you provided.

2.

About the ML Models (lines 272-275) I am not expert in that one the authors used, but I guess there other based on different AI algorithms such as artificial neural networks, which include

several statistical fitness criteria to obtain the best train Set R to validate the predictions. For example, Cross-Validation which split the data into subgroups that can be used for training, other for test and finally, others unseen for the model, to validate. Have you split your data in those groups or used all for training and/or testing? May be you can used other (there are several) AI algorithms for model and in silico validate your model.

Also, did you carried out any statistical method (such as ANOVA) to verify that there are not statistical differences between the experimental and predicted values in order to assess the model accuracy? (see lines 276-279). If done, please include those data in the manuscript. In this sense, I miss a table with the training parameters used for the model (currently not included).

I suggest to explore the use of other ML models, based in Artificial Neural Networks, to model these results and establish what ion was/were key (P or N or K) in the parameters detemined (cellulose, lignin, lipids and so on, content).

[Author's Response] Thank you this is an excellent suggestion. We had used the ANN for classification of mineral types in a recent publication for your reference (Hao, et al. Sci Rep 8, 2552 (2018). <https://doi.org/10.1038/s41598-018-20365-6>). Regression models are preferred in this study because we would like to better quantify the N/P concentrations so we could validate the model more conveniently. Exploration of other ML models is definitely part of our future work.

The rest of the papers is excellent and worthy to be published.

[Author's Response] Thank you for your kind words.

Minor issues:

Line 117. Please use SI units instead "inch" (such as centimeters or milimeters)

[Author's Response] Thank you, we've updated the manuscript to include values with the SI units.

Reviewer #3 (Remarks to the Author):

The manuscript presents a study on the prediction of P content using spectroscopy and machine learning (Principal Component Regression). This is a novel and solid scientific work and I have only two minor suggestions. First, it would be useful to include in the introduction some background regarding the use of Machine Learning techniques for nutritional analysis. Investigations on the subject are aplenty and have yielded some very interesting results. Since this is not a review article, I would not expect an in-depth analysis, but some general comments on the state of the art would be useful. Second, it would be interesting if the authors could speculate a little about the generality of the results obtained. Can we expect that the results would be the same if the study was carried out in areas with different climate and soil characteristics?

[Author's Response] Thank you for your review and great suggestions. We've updated the manuscript to include a short paragraph to introduce the use of ML for nutrition studies. We would expect the same approach could be used in areas with different climate and soil conditions, however, in parallel with a carefully designed laboratory-based experiment. The current study we presented here will need to include more variable climate conditions to make the methodology more generalizable. We are very interested in collaborating with relevant experts around the world to find ways to optimize this approach.

Reviewers' comments:

Reviewer #1 (Remarks to the Author):

Please provide a file with marked changes (track changes) in it for your future publications which makes it easier for the reviewer.

The revised manuscript is satisfactory which can be published in Communications Biology.

Reviewer #2 (Remarks to the Author):

From my side, the manuscript can be now accepted.

Reviewer #3 (Remarks to the Author):

My suggestions were properly addressed and I do not have any further comments.

Reviewer #4 (Remarks to the Author):

This manuscript is well written and contains interesting analysis of the role of P in switchgrass-plantations. However, the exact data used for the ML-method is not very well described, neither is the validation procedure of the method. My concern about the method is that the model fit to the cross-validation data seems to give a perfect fit – why is that? As the data used in the model training and validation are measurements, they probably contain some measurement error. Now it seems that the model can predict also those errors (=overfit)?

Detailed comments:

Abstract: In the abstract, the emphasis of this study seems to be the development of ML-model (actually an existing model type, PCoR, is used here, the novelty is in the data with which it is trained?). However, the analysis includes also laboratory & field sample based results of different nutrients, and Pi versus Po analysis. It should be more clearly clarified, what is the actual novel method of this study (referred in the manuscript as "our method").

L81: Linear modes -> linear models

L111-113: What was the value of N when varying Pi, and what was the value of Pi when varying N? What were the sample sizes for each set of set?

L124: What is meant by plot here - the two test locations?

L190-193: What exactly were the independent and dependent variables in PCoR:s here? (The authors explain this in their previous response to reviewers, but this remains still unclear in the manuscript.) How much correlation were among independent variables (which often are not independent but highly correlated with each other...)? Why the size 6000 for random samples was chosen?

L194: What is meant by iterating through extracted PCs`? How was the iteration performed?

L194-196: Why does the cross-validation error get this small, even to zero if more PC:s are used? It seems that there is a perfect fit for the model? Perfect fit of a model is an alarming sign of possible overfit of the model (overfitting can occur, when the number of used variables is high compared to the training set size), and thus it should be carefully analyzed why it is happening.

What were the test set sizes and validation set sizes in the 10-fold cross-validation procedure, explain that in the manuscript. It would be good to see a figure of the predicted values and the corresponding response values from the measurements (used in cross-validation).

From the previous response of the authors, it seems that the training set size in each CV-step is about 210 sample points (90% of the data from 78 independent sample points with 3 replicants of each) (?) with measured dependent variables (P or N) and a total of 313 frequency bands used as independent variables? Thus, the number of 44 PC:s used seems quite high compared to the training set size of 210, or only about 70 (about 90% of 78 sample points) independent sample points.

L265: To what method do the authors refer here "..., but our method...?"

L283-284: R2 is about one -> the PCR-model is assumed to give exact predictions with no error at all?

L339-345: This paragraph discusses the results of Pi and Po together. However, it is in the chapter of ML-results, which is a little confusing, as the model does not (?) give the Pi and Po predictions separately. It should be clearly stated here, that authors compare the results & conclusions of ML-model based predictions to measurement based analysis.

L371: What is meant by "differential" here, clarify.

L384: The characterization method -> what method is this?

General: How well the ML-approach used here would work in a real-world situation, with remotely sensed spectra (L76-89 indicate that the AI-methods similar to this are used with remotely sensed data) and varying conditions? If I understood correctly, in this study no remotely sensed measurements were used, but spectral features of younger leaf samples instead?

Reviewers' comments:

We appreciate the great comments from all the reviewers! We take our study and the reviewers' comments seriously and we are grateful for these constructive suggestions which have greatly improved our work. In this response, we will only reply to the outstanding comments from reviewer #4.

Reviewer #1 (Remarks to the Author):

Please provide a file with marked changes (track changes) in it for your future publications which makes it easier for the reviewer.

The revised manuscript is satisfactory which can be published in Communications Biology.

Reviewer #2 (Remarks to the Author):

From my side, the manuscript can be now accepted.

Reviewer #3 (Remarks to the Author):

My suggestions were properly addressed and I do not have any further comments.

Reviewer #4 (Remarks to the Author):

This manuscript is well written and contains interesting analysis of the role of P in switchgrass-plantations. However, the exact data used for the ML-method is not very well described, neither is the validation procedure of the method. My concern about the method is that the model fit to the cross-validation data seems to give a perfect fit – why is that? As the data used in the model training and validation are measurements, they probably contain some measurement error. Now it seems that the model can predict also those errors (=overfit)?

[Author's Response] We appreciate the comments from the reviewer. Obviously the reviewer had raised important questions and concerns though we had tried our best to answer some aspects of the similar concerns from reviewer #1. We are very glad to have this opportunity to address this “perfect fit” concern with more details in the response to comment #7.

Detailed comments:

1.

Abstract: In the abstract, the emphasis of this study seems to be the development of ML-model (actually an existing model type, PCoR, is used here, the novelty is in the data with which it is trained?). However, the analysis includes also laboratory & field sample based results of different nutrients, and Pi versus Po analysis. It should be more clearly clarified, what is the actual novel method of this study (referred in the manuscript as “our method”).

[Author's Response] Thank you for your comments. In our view, the manuscript describes a new advance in our ability to understand dynamic soil phosphorus (P) availability, along with P speciation and allocation strategies of switchgrass cultivated under nutrient limitation such as found in marginal soils. This is the main reason why we sent the study to Communications Biology. Meanwhile, we have developed a machine learning method to predict soil P availability to plants using infrared spectral data from leaf material. Rather than just assessing P content in leaves we use the biochemical response of the plant to soil P availability as a sensor of plant available P throughout the growing season. This may sound intuitive, but solid evidence like we present here is rarely produced. We look forward to your further thoughts on this.

2.

L81: Linear modes -> linear models

[Author's Response] Apologies for the typo. Thank you for spotting it. We have revised it in the resubmitted manuscript.

3.

L111-113: What was the value of N when varying Pi, and what was the value of Pi when varying N? What were the sample sizes for each set of set?

[Author's Response] We have updated with this information: A total of 78 plants were watered daily to field capacity during a 4-week growth period, half of which with a nutrient solution containing 1, 10, 30, 150 or 500 μM of Pi (with optimal 6 mM of N), supplied as KH_2PO_4 , and the other half with 0.01, 0.1, 0.3, 1.5 or 6 mM of N (with 500 μM of P), supplied as KNO_3 . There were 9 replicates for each condition, except that three of the plants grown with $\text{Pi}=1 \mu\text{M}$ had died before the harvest, which were not included in the analysis ... Three replicates of each sample were measured with FTIR.

4.

L124: What is meant by plot here - the two test locations?

[Author's Response] Yes.

5.

L190-193: What exactly were the independent and dependent variables in PCoR:s here? (The authors explain this in their previous response to reviewers, but this remains still unclear in the manuscript.) How much correlation were among independent variables (which often are not independent but highly correlated with each other...)? Why the size 6000 for random samples was chosen?

[Author's Response] We updated the lines 194-195 to make it more clear what were the independent and dependent variables. The independent variables were infrared spectra of the leaves as shown in Figure S9. They are usually not highly correlated in this type of plant material. The sampling size of 6000 was arbitrarily chosen based on the law of large numbers.

Again, each spectrum contains 313 frequency bands (313 independent variables). Our sample size, as you have commented later, was admittedly not large. In order to extract eigenvalues from a matrix, the matrix has to be further populated to contain at least 313 samples, which is the main reason here for the use of a bootstrapping method. A size of 600 in our test produces a similar learning curve and similar results as with the size of 6000.

6.

L194: What is meant by iterating through extracted PCs`? How was the iteration performed?

[Author's Response] Simply put, PCoR is a linear regression model using principal components as regressors (selected from eigenvalues of the data matrix). The details can be found in the references

[40] and [43]. During each round of a 10-fold cross-validation, a different number of PCs was used to “fit” to the data, and a best-fit MSE_P was derived. After these iterations, we built a model which contained submodels with different numbers of PCs, along with the evaluation of prediction accuracy. A learning curve then was obtained to evaluate the model built from the lab experiments.

7.

L194-196: Why does the cross-validation error get this small, even to zero if more PC:s are used? It seems that there is a perfect fit for the model? Perfect fit of a model is an alarming sign of possible overfit of the model (overfitting can occur, when the number of used variables is high compared to the training set size), and thus it should be carefully analyzed why it is happening.

What were the test set sizes and validation set sizes in the 10-fold cross-validation procedure, explain that in the manuscript. It would be good to see a figure of the predicted values and the corresponding response values from the measurements (used in cross-validation).

From the previous response of the authors, it seems that the training set size in each CV-step is about 210 sample points (90% of the data from 78 independent sample points with 3 replicants of each) (?) with measured dependent variables (P or N) and a total of 313 frequency bands used as independent variables? Thus, the number of 44 PC:s used seems quite high compared to the training set size of 210, or only about 70 (about 90% of 78 sample points) independent sample points.

[Author’s Response] First, the 10-fold cross-validation was performed on the lab-based experiment, with near perfect predictors, i.e. the N/P concentrations of the growth medium which was supplied on the daily basis, with 9 replicates for each which further reduced the error. The percentage error of the analytical balance was around 10^{-4} % depending on the molecular weight. The plant leaves were freeze-dried right after harvest and homogenized in liquid nitrogen. Yes, it took a lot of effort to minimize analytical error!

Second, the infrared data are extremely sensitive to the chemical profile of leaf tissues, which are extremely sensitive to the nutrient inputs. We used over 300 frequency bands in the mid-IR signature region and three replicates for each sample (so a total of $3 \times 78 = 234$ samples). The rich input information and the high sensitivity of the leaf as a “sensor” resulted in the high R² value achieved by our “simple” linear machine learning model, which validated our ML approach.

Third, one purpose of using cross-validation is to avoid overfitting. A typical learning curve of supervised learning (please compare to ours in Supplementary Figure S5) from a rigorous cross-validation would give you information of whether the model is underfit or overfit. See the following diagram of a typical learning curve, compared with one from a PCoR model built from a randomly-generated dataset. It’s very unlikely to overfit data with linear models.

Finally the small training error was partly attributable to the use of bootstrapping and the law of large numbers, which further improved the prediction accuracy.

8

L265: To what method do the authors refer here “..., but our method...”?

[Author’s Response] Here it means the spectroscopic method. Thank you for pointing that out. We have revised it accordingly.

9

L283-284: R² is about one -> the PCR-model is assumed to give exact predictions with no error at all?

[Author’s Response] That learning curve represents the training process with the goal to build an optimized mode. In our view, there was nothing wrong for a model accuracy approaching to be perfect. However, here are the caveats. Although the R² obtained from the lab-experiment had graded the model with an A or A+, it didn’t imply that the R² of prediction on the field experiment would be equally perfect. That is why we spent time to further validate the field prediction with root data and a second method, as shown in Fig S6, S7 and S8.

10

L339-345: This paragraph discusses the results of Pi and Po together. However, it is in the chapter of ML-results, which is a little confusing, as the model does not (?) give the Pi and Po predictions separately. It should be clearly stated here, that authors compare the results & conclusions of ML-model based predictions to measurement based analysis.

[Author's Response] We believe our model predicted the overall P inputs in the root zone, given that our dependent variables in the model were P concentrations in the growth medium, thus its prediction didn't discern P_i from P_o . However, the discussion of P_i and P_o (distinguishable from the spectra) was meant to provide valuable context for the changes in total P.

Specifically we state that *"The increase in P concentration in the late season in roots and the speciation of P_o and P_i in leaves indicate that switchgrass plants increase P mobilization and storage during shoot senescence, as discussed above. We further explored this point and observed a clear association between second-year biomass production and the ratio of P_i/P_o in the shoots at senescence (Supplementary Figure S8). This suggested that effective reallocation of P (i.e., a higher P_i/P_o ratio in the late season) during senescence may contribute to the increased total biomass observed in second-year growth ($p = 4.4 \times 10^{-16}$)."* We believe this text provides accurate context for how the interplay between P_i and P_o concentrations in the plant tissue can be used to infer P remobilization associated with changes in total P that was predicted by the ML model.

11

L371: What is meant by "differential" here, clarify.

[Author's Response] We use it as defined in Merriam Webster dictionary "of, relating to, or constituting a difference", in this case differences "in P reallocation within leaf tissue" and differences in the "recovery of total P in root tissue late in the growing season".

12

L384: The characterization method -> what method is this?

[Author's Response] The characterization method referred to the learning-assisted spectroscopic method to predict the differential P reallocation, which could be potentially used to predict the second year feedstock yield and quality.

13

General: How well the ML-approach used here would work in a real-world situation, with remotely sensed spectra (L76-89 indicate that the AI-methods similar to this are used with remotely sensed data) and varying conditions? If I understood correctly, in this study no remotely sensed measurements were used, but spectral features of younger leaf samples instead?

[Author's Response] We believe that a similar method could be developed to better interpret and correlate remote sensing data to nutrient analysis at larger scales. We have cited some references

doing that, and we hope our manuscript may help improve those models also. We thank the reviewer again for the meticulous comments.

Reviewers' comments:

Reviewer #4 (Remarks to the Author):

I thank the authors for their thorough response and clarification of the model structure and used data. I think the details are now better explained in the manuscript.

However, I still find it problematic to trust the model that seems so over-optimistic - perfect fit of the model in a case where the measurements are derived from living plants and infrared data, which both likely contain some variability? Can these data sources really be combined such that the model gives perfect predictions?

Also linear models can give over-fitting models, when the number of independent variables is close to number of sample points in the training data. Also, perfect fit can be achieved within the training set, and if the validation set is not independent of the training set, it will also give nearly perfect fit.

In this study, there were 78 independent plants with nutrient measurements, i.e. 78 independent response variables. Three replicates of each mid-IR signature region per plant were used in the sample, i.e. there were three replicates of each plant in the response set (?). Thus, there are only 78 independent sample points in the data used in the 10-fold cross-validation procedure.

How did the authors divide this set of 234 sample points in the training set and validation set for each of the 10-fold cross-validation repetition? Can the same plant be in both sets (e.g. one replicant in the validation set, two replicants in the training set)? Or is the validation procedure performed correctly such, that all the plant level data is set either in the validation, or in the training set, in each of the 10 repetitions?

This procedure should be explained with details also in the manuscript.

Reviewer #4 (Remarks to the Author):

I thank the authors for their thorough response and clarification of the model structure and used data. I think the details are now better explained in the manuscript.

[Author Response] Thank you for the comments. We appreciate your further thoughts on this manuscript. We have participated in transparent peer review, so we believe what we have discussed here will greatly help the readers to better understand the data analysis process once the paper is published. We will make our code and data available to the public for further scrutiny.

However, I still find it problematic to trust the model that seems so over-optimistic - perfect fit of the model in a case where the measurements are derived from living plants and infrared data, which both likely contain some variability? Can these data sources they really be combined such that the model gives perfect predictions?

Also linear models can give over-fitting models, when the number of independent variables is close to number of sample points in the training data. Also, perfect fit can be achieved within the training set, and if the validation set is not independent of the training set, it will also give nearly perfect fit.

In this study, there were 78 independent plants with nutrient measurements, i.e. 78 independent response variables. Three replicates of each mid-IR signature region per plant were used in the sample, i.e. there were three replicates of each plant in the response set (?). Thus, there are only 78 independent sample points in the data used in the 10-fold cross-validation procedure.

How did the authors divide this set of 234 sample points in the training set and validation set for each of the 10-fold cross-validation repetition? Can the same plant in both sets (e.g. one replicant in the validation set, two replicants in the training set)? Or is the validation procedure performed correctly such, that all the plant level data is set either in the validation, or in the training set, in each of the 10 repetitions?

This procedure should be explained with details also in the manuscript.

[Author Response] Please note that in our last response letter, we have stated that the merits of this paper are not limited within this machine learning aspect, so we hope you consider all the merits of the manuscript before you make your recommendation. This is an important reason of why we did not provide paragraphs of explanation about how the model was trained in the manuscript.

However, we did provide quite detailed information in our last response letter mainly regarding your comment #7. Your current concern had been similarly raised in the comments #4, #6 and #8 from reviewer #1 in the first round of review, who later accepted and agreed with our explanation. We still believe these responses are valid and they support our findings in our manuscript. We want to reiterate the following two opinions to address your concern about the potential overfitting of our data with a linear model based on principal component regression and the allocation of our samples between training and validation sets in the development of our model.

1. **The 10-fold cross-validation process we used is standard for model training and evaluation.** We simply do not see any evidence of overfitting in our case, because R2 was not reduced by increasing the number of PCs (Fig. S5). As stated in [Peter Flach, Machine Learning: The Art and Science of Algorithms that Make Sense of Data, Cambridge 2014, page 19], “If overfitting occurs, the test set performance will be considerably lower than the training set performance”. Furthermore, the author states “even if we select the test instances randomly from the data, every once in a while we may get lucky ... or not lucky ... In practice this train-set split is therefore repeated in a process called cross-validation ...”. The cross-validation process is exactly designed to prevent overfitting from happening.
2. **By definition, a k-fold cross-validation starts with a random partition of data in k (k=10 in our case) parts or folds (segments in Ref [43]).** We train the model on the k-1 folds and then validate with the one-fold. This process is then repeated k times until each fold has been used for testing once. We did not do this manually or create our own code. We want to emphasize “random partition” here as it has directly answered questions about the allocation of data points.

Our text in the methodology section states: *“The optimal number of components for the PCoR model was selected by iterating through extracted PCs to obtain the lowest cross-validation error from a 10-fold cross-validation (mean squared error of prediction, MSEPE < 0.01)”*. This is a precise statement of our training process, and the 10-fold cross-validation refers to the random allocation of 90% of data for training and 10% for validation. We had provided reference [43] for further detailed information regarding this data allocation in the *pcr* function with a “CV” option we used in the *pls* R package. Here we cite the process described by this reference in the following quoted text.

“Cross-validation, commonly used to determine the optimal number of components to take into account, is controlled by the validation argument in the modelling functions (mvr, pls and pcr). The default value is “none”. Supplying a value of “CV” or “LOO” will cause the modelling procedure to call mvrCv to perform cross-validation; “LOO” provides leave-one-out cross-validation, whereas “CV” divides the data into segments. Default is to use ten segments, randomly selected, but also segments of consecutive objects or interleaved segments (sometimes also referred to as ‘Venetian blinds’) are possible through the use of the argument segment.type. One can also specify the segments explicitly with the argument segments; see ?mvrCv for details.

When validation is performed in this way, the model will contain an element comprising information on the out-of-bag predictions (in the form of predicted values, as well as MSEPE and R2 values). As a reference, the MSEPE error using no components at all is calculated as well. The validation results can be visualised using the plotype = “validation” argument of the standard plotting function. An example is shown in Figure 2 for the gasoline data; typically, one would select a number of components after which the cross-validation error does not show a significant decrease. Unfortunately, no generally applicable tests are available, so the decision on how many components to retain will always be subjective to some extent.”

After we obtained a satisfactory model training outcome via cross-validation, we then applied the model to interpret the fully-independent field data to estimate the phosphorus availability in the rhizosphere. This was then further compared against complementary measurements of plant P dynamics using additional independent data. We therefore believe we have multiple solid lines of evidence to support our conclusions.

REVIEWERS' COMMENTS:

Reviewer #4 (Remarks to the Author):

Although the concept of independent validation sets doesn't seem to be exactly clear in this study, I find the authors' explanations sufficient, especially as the used ML approach doesn't play a major role in this study. Thus, I think the manuscript can be accepted for publication.